# Learning Dynamics of Deep Networks Admit Low-Rank Tensor Descriptions

**Christopher H. Stock**[1,*] **Alex H. Williams**[1,*] **Madhu S. Advani**[2], **Andrew M. Saxe**[2], **Surya Ganguli**[1]
[1]Stanford University, Stanford, CA, USA; [2]Harvard University, Cambridge, MA, USA
{chstock,ahwillia,sganguli}@stanford.edu, {madvani,asaxe}@fas.harvard.edu

## Abstract

Deep feedforward neural networks are associated with complicated, nonconvex objective functions. Yet, simple optimization algorithms can identify parameters that generalize well to held-out data. We currently lack detailed descriptions of this learning process, even on a qualitative level. We propose a simple tensor decomposition model to study how hidden representations evolve over learning. This approach precisely extracts the correct dynamics of learning in linear networks, which admit closed form solutions. On deep, nonlinear architectures performing image classification (CIFAR-10), we find empirically that a low-rank tensor model can explain a large fraction of variance while extracting meaningful features, such as stage-like learning and selectivity to inputs.

## 1 Introduction and proposed tensor model

Deep networks can be difficult to optimize, yet carefully designed architectures and clever initialization and normalization strategies often yield state of the art performance on a variety of tasks. Prior work has characterized deep network optimization by visualizing 1D or 2D slices of the objective function (Goodfellow et al., 2015; Li et al., 2017). While these studies provide a useful glimpse into learning dynamics, they summarize the extremely rich and complex input-output functions of deep networks as a single number (the loss). Other investigations into the optimization landscape of deep networks include (Dauphin et al., 2014; Choromanska et al., 2015; Guo et al., 2015).

To empirically study learning dynamics, we record the activations of $N$ units, in response to $M$ test inputs, at $T$ instances over training. These data are naturally expressed as a $N \times M \times T$ tensor, which we denote $\boldsymbol{\mathcal{X}}$. In modern settings, $\boldsymbol{\mathcal{X}}$ may be very large and contain complex structure. However, we hypothesized that a simple model may still explain a significant fraction of variance and yield insight into cases of theoretical and practical interest. In particular, we sought to approximate $\boldsymbol{\mathcal{X}}$ as a low-rank tensor, $\widehat{\boldsymbol{\mathcal{X}}}$, which can be compactly expressed by its canonical polyadic (CP) decomposition:

$$[\widehat{\boldsymbol{\mathcal{X}}}]_{nmt} = \sum_{r=1}^{R} u_n^r v_m^r w_t^r . \tag{1}$$

The rank of $\widehat{\boldsymbol{\mathcal{X}}}$ is $R$, which represents the number of components in the model. Each component, indexed by $r$, consists of a triplet of vectors: $\mathbf{u}^r \in \mathbb{R}^N$, $\mathbf{v}^r \in \mathbb{R}^M$, and $\mathbf{w}^r \in \mathbb{R}^T$, which we call *factors* in analogy to learned features from matrix factorization models. Furthermore, as with matrix factorization models, these factors have intuitive interpretations: each vector $\mathbf{u}^r$ describes a pattern of activity across the $N$ neurons in a layer, each $\mathbf{v}^r$ indicates which images in the test set elicit this pattern of activity, and each $\mathbf{w}^r$ vector contains a learning curve for this neural representation. Thus, we refer to $\mathbf{w}^r$ as *learning factors*, $\mathbf{v}^r$ as *input factors*, and $\mathbf{u}^r$ as *neuron factors*. We fit these factors using alternating least squares (with multiple random initializations), which is a standard method for this model class (Carroll & Chang, 1970; see Kolda & Bader, 2009, for review; see also Williams et al., 2017 for a similar application to neuroscience data).

We emphasize two crucial features of the CP decomposition model:

---

*Equal contribution.

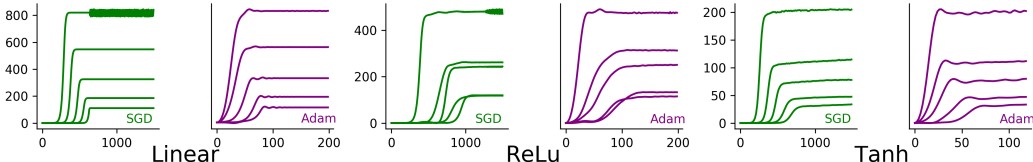

Figure 1: The growth of learning modes in linear, ReLu, and tanh networks with one hidden layer, trained to zero test error to replicate a teacher network of the same architecture, with initialization near the origin. For each architecture, we analyzed learning dynamics under gradient descent (l.r.=0.3) and Adam (l.r.=0.003). Each plot shows the learning factors $\{\mathbf{w}^r\}_{r=1}^5$ from a rank-5 tensor decomposition, which in every case accounted for at least 85% of the variance. The $x$-axis is training epoch, and the $y$-axis is strength of learning mode (a.u.).

- First, unlike subspace identification methods like PCA or CCA (which have previously been applied to deep networks; Raghu et al., 2017) the CP tensor model can often recover non-orthogonal features (Kruskal, 1977). This is particularly likely if we restrict $\mathbf{u}^r$, $\mathbf{v}^r$, and $\mathbf{w}^r$ to contain nonnegative entries (Qi et al., 2016), which is a natural constraint for networks with rectified linear units.

- Second, hypothesizing low-rank structure in higher-order tensors is a much stronger statement than in the matrix case. An $N \times N \times N$ tensor contains $N^3$ entries, while the CP model has only $O(RN)$ parameters. This massive difference in dimensionality implies that a low-rank tensor dramatically simplifies our understanding of large-scale data.

## 2 LINEAR NETWORKS ADMIT PRECISE LOW-RANK DECOMPOSITIONS

Learning dynamics have been most rigorously studied in linear networks, which admit closed form solutions under gradient descent dynamics, whitened input statistics, and initialization near the origin. In this regime, Saxe et al. (2014) showed that a network with a bottleneck layer of $R$ neurons learns to match the top $R$ singular vectors of the input-output covariance matrix $\Sigma^{xy}$. Furthermore, learning dynamics *decouple* in the basis of these singular vectors, and the rate at which each pair of singular vectors is learned is inversely proportional to the singular value.

We point out that decoupled learning dynamics analytically derived by Saxe et al. (2014) formally map onto a low-rank tensor. Specifically, the left and right singular vectors of $\Sigma^{xy}$ respectively map onto the subspace spanned by the input factors, $\mathbf{v}^r$, and the neuron factors, $\mathbf{u}^r$. The learning factors, $\mathbf{w}^r$, map onto the sigmoidal learning curves that were analytically derived by Saxe et al. (2014). To demonstrate this equivalence, we generated data from a low-rank linear teacher network with $M$ inputs and $M$ outputs ($M = 100$), and we trained a linear student network with one hidden layer of $M$ neurons to zero test error under squared error loss. We collected data into a tensor $\boldsymbol{\mathcal{X}}$ as described in section 1, and recovered the exact results of Saxe et al. (2014) in a purely unsupervised manner (Fig. 1, far left panel; input and neuron factors not shown).

While linear networks can provide important insights, there is a large gap between this analytically tractable setting and modern machine learning applications. However, since the tensor decomposition model is entirely agnostic to the architecture and task, we wondered whether this framework could empirically describe learning in cases where the assumptions of Saxe et al. (2014) were broken. We used this technique to show that *nonlinear* networks with one hidden layer trained on *nonlinear* teachers exhibit qualitatively similar low-rank dynamics to the linear case, under the learning dynamics of both gradient descent and Adam (Fig. 1). In these settings, very low-rank tensor decompositions were routinely able to capture over 85% of the variance of the original learning tensor.

## 3 CATEGORY-SPECIFIC LEARNING DYNAMICS IN A DEEP CONVNET

We next tested the CP decomposition model in a far more challenging setting: a convolutional network with 8 hidden layers trained on CIFAR-10 images (Kaur, 2017). To simplify our analysis and reduce memory requirements, we recorded the maximal activation for each convolutional filter

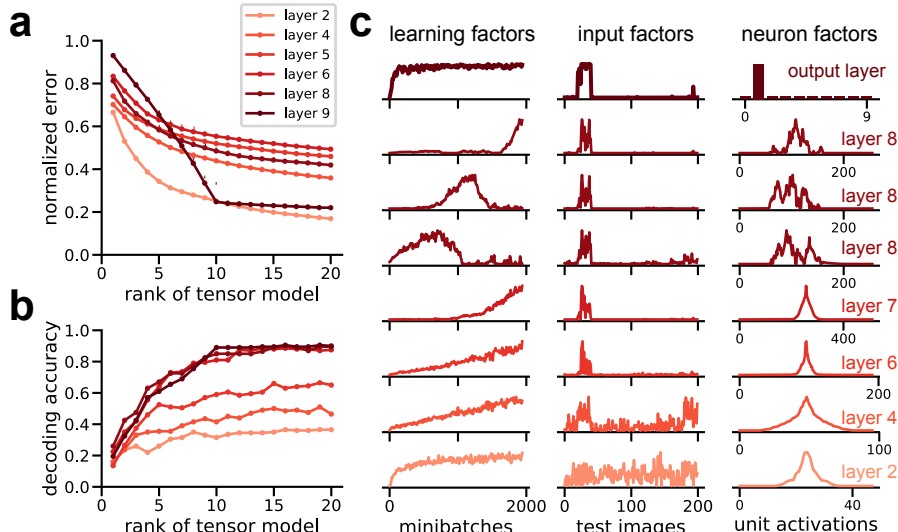

Figure 2: **(a)** Reconstruction error vs. number of tensor components. **(b)** Accuracy of a linear classifier trained on tensor components. **(c)** Representative learning factors ($\mathbf{w}^r$, left), input factors ($\mathbf{v}^r$, middle), and neuron factors ($\mathbf{u}^r$, right) discovered on various network layers (color-coded).

into the data tensor, thus capturing a position-invariant measure of activation. We fit nonnegative decompositions to data collected from each layer on 200 test images (20 images per category).

These models described a surprisingly large fraction of variance in this dataset. Figure 2a shows the normalized reconstruction error ($\|\widehat{\boldsymbol{\mathcal{X}}} - \boldsymbol{\mathcal{X}}\|_F / \|\boldsymbol{\mathcal{X}}\|_F$) for several layers as a function of tensor rank, $R$. Taking layer 8 (the final hidden layer) as an example, a rank-20 model explained ~60% of the (uncentered) variance in the data, roughly a 2-fold improvement over a rank-1 model. Layer 9 (the output layer) is even more cleanly described by a model with 10 components, corresponding to the 10 image categories, as indicated by a suggestive kink in the normalized error at $R = 10$.

Most importantly, tensor decompositions empirically identified factors that were human-interpretable, task-relevant and of practical interest. For example, using input factors, $\mathbf{v}^r$, from any of the final four layers, we were able to infer the labels of the test images by a multiclass logistic regressor (~90% accuracy, 10-fold cross-validated). This level of accuracy was comparable to (and slightly exceeded) the accuracy of the network on the full test set (~84%).

The tensor factors themselves reveal interesting trends in the data. To provide a digestible and representative sample, we chose 8 components (rows of Fig. 2c) across several layers of the network that showed some selectivity for "automobile" images (test images 20-40). Similar results were obtained for other image classes (not shown). In deep layers (upper rows), the input factors (middle column) were sparse and selective for automobiles, while earlier layers were less selective. The learning dynamics varied strongly across layers: learning in the final layer increased rapidly before saturating while layers 4 and 6 ramped slowly over learning. Deeper layers, such as layer 8 (Fig 2c, blue), exhibited non-monotonic learning curves, suggesting shifting representations for the same image class. Finally, the neuron factors (right column, sorted to illustrate sparsity) suggest that learning in earlier layers tends to involve broader populations of neurons, whereas learning in deeper layers is constrained to smaller, more sparse populations.

The framework we propose makes few assumptions about the computational task and network model and thus may illuminate learning in a variety of settings, including recurrent architectures. For example, we will investigate the extent to which hidden units that participate together in learning (as identified by neuron factors) exhibit shared patterns in their input and output weights. We will also scale our experiments up to larger-scale networks and test sets by incorporating recently developed randomized algorithms for tensor decomposition (Erichson et al., 2017; Battaglino et al., 2017).

ACKNOWLEDGMENTS

C.H.S. thanks the Blavatnik Family Foundation; A.H.W. thanks the U.S. Dept. of Energy Computational Science Graduate Fellowship (CSGF) program; M.S.A. and A.M.S. thank the Swartz Program in Theoretical Neuroscience at Harvard; and S.G. thanks the Simons, McKnight, James S. McDonnell, and Burroughs Wellcome Foundations and the Office of Naval Research for support.

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
