# OpenReview forum: "Learning Dynamics of Deep Networks Admit Low-Rank Tensor Descriptions"
_ICLR.cc/2018/Workshop — Reject_

### Official Review · AnonReviewer2 · 2018-03-06
**Interesting use of tensor decomposition**

**Rating:** 5
**Confidence:** 4

**Review:**

In this paper, the dynamics of how neural networks are learned is analyzed. For every training instance, they record activations of units for several test instances, which yields a 3-order tensor (activation x test x training). To reduce the data complexity, CP decomposition is used. Interestingly, CP decomposition can be also helpful to interpret the tensor.

The experimental results are interesting. However, the results are mostly evaluated in qualitative ways and there is no rigorous evidence that the proposed analysis is really useful. Also, it isn't described how we can use the proposed method for real use cases. For example, the method is possibly useful to debug the learning process of DNNs (e.g. detecting the wrong choice of learning rate). Adding such discussion will enhance the impact of the paper.

Pros:
- The idea of using tensor decomposition to analyze learning dynamics is unique.

Cons:
- Experimental results are qualitative.
- Not ready to real problems.

---

### Official Review · AnonReviewer1 · 2018-03-09

[review text omitted: it was posted to a different submission]

---

### Official Review · AnonReviewer3 · 2018-03-19
**This paper provides a new perspective to study the learning dynamics in linear networks.**

**Rating:** 6
**Confidence:** 4

**Review:**

This paper proposes a CP tensor decomposition model to study the learning dynamics in linear networks. The authors show that this low-rank tensor model can empirically explain a large fraction of variance while extracting meaningful features on CIFAR-10 dataset.

Although this is a novel insight to represent the neural networks by CP decomposition factors, this paper also has the following questions:
1. The authors only show these factors that can interpret the neural networks intuitively, but the theoretical  causes are not exlored (although this may be difficult).

2. In Fig 2(a), why does the ‘layer 2’ instead of the 'output layer' have the lowest error ? Or why is there such a rule: the error first increases first and then decreases as the layer goes deeper?

3. Lack of comparative experiment. The paper only analyzes the performance of the proposed method w.r.t some hyper-parameters, e.g., rank R, test images and so on. How about  the performance of other  exsiting methods?

---

### Decision · Program_Chairs · 2018-03-20
**ICLR 2018 Workshop Acceptance Decision**

**Decision:**

Reject

**Comment:**

Based on the reviews, this paper has not been accepted for presentation at the ICLR workshop. However, the conversation and updates can continue to appear here on OpenReview.